# The child's pantheon: Children's hierarchical belief structure in real and non-real figures

**Rohan Kapitány**[1,2]*, **Nicole Nelson**[3], **Emily R. R. Burdett**[2,4], **Thalia R. Goldstein**[5]

**1** School of Psychology, Keele University, Keele, Staffordshire, England, United Kingdom, **2** School of Anthropology and Museum Ethnology, University of Oxford, Oxford, England, United Kingdom, **3** School of Psychology, The University of Queensland, Brisbane, Australia, **4** The University of Nottingham, Nottingham, England, United Kingdom, **5** Department of Psychology, George Mason University, Fairfax, Virginia, United States of America

* r.f.kapitany@keele.ac.uk

**Data Availability Statement:** Our full data are available at https://osf.io/wurxy/.

**Funding:** This research was supported by funding from Horizon 2020 Excellent Science Grant provided by the ERC (Ritual Modes, No. 694986).

## Abstract

To what extent do children believe in real, unreal, natural and supernatural figures relative to each other, and to what extent are features of culture responsible for belief? Are some figures, like Santa Claus or an alien, perceived as more real than figures like Princess Elsa or a unicorn? We categorized 13 figures into five a priori categories based on 1) whether children receive direct evidence of the figure's existence, 2) whether children receive indirect evidence of the figure's existence, 3) whether the figure was associated with culture-specific rituals or norms, and 4) whether the figure was explicitly presented as fictional. We anticipated that the categories would be endorsed in the following order: 'Real People' (*a person known to the child*, *The Wiggles*), 'Cultural Figures' (*Santa Claus*, *The Easter Bunny*, *The Tooth Fairy*), 'Ambiguous Figures' (*Dinosaurs*, *Aliens*), 'Mythical Figures' (*unicorns*, *ghosts*, *dragons*), and 'Fictional Figures' (*Spongebob Squarepants*, *Princess Elsa*, *Peter Pan*). In total, we analysed responses from 176 children (aged 2–11 years) and 56 adults for 'how real' they believed 13 individual figures were (95 children were examined online by their parents, and 81 children were examined by trained research assistants). A cluster analysis, based exclusively on children's 'realness' scores, revealed a structure supporting our hypotheses, and multilevel regressions revealed a sensible hierarchy of endorsement with differing developmental trajectories for each category of figures. We advance the argument that cultural rituals are a special form of testimony that influences children's reality/fantasy distinctions, and that rituals and norms for 'Cultural Figures' are a powerful and under-researched factor in generating and sustaining a child's endorsement for a figure's reality status. All our data and materials are publically available at https://osf.io/wurxy/.

## Introduction

Children's understanding of the real and unreal tends to be largely nuanced and accurate [1–3]. By the age of three children can distinguish between veridical, imagined, and pretend entities [4,5] and between superficial and actual features of objects [6]. Even when children are incorrect, it is often due to systematic cultural factors rather than their own idiosyncrasies or

**Competing interests:** The authors have declared that no competing interests exist.

cognitive abilities [7–9]. And in such cases children still show accuracy on the properties and limits on the content of supernatural minds, suggesting a nuanced understanding of what such figures can and cannot know [3,10,11]. And yet we have little idea how various kinds of non-real figures are evaluated *relative* to each other: are different types of supernatural figures endorsed with more or less confidence, and can children sensibly admit ambiguity within the spectrum of their beliefs?

Not all supernatural figures are created equal. Some are concerned in particular domains of life or behavior, while some are not [12]. Some act upon the world in direct ways, while others do not interact at all; the presence of some figures is conditional, such that a set of behaviors must be followed prior to, or in response to, the supernatural figure's arrival. The latter is most true for the most famous of the child's pantheon—Santa Claus, The Easter Bunny, and The Tooth Fairy—who, somewhat uniquely in the western/European canon, require those who believe to perform certain culturally sanctioned ritualistic actions. And while traditional forms of testimony certainly contribute to belief, we argue that it may not be a coincidence the supernatural figures commonly endorsed as real by children are the same figures who require children to act in particular, culturally sanctioned ways.

Woolley and Ghossainy [13] argue that three factors can persuade a child of a figure's veridicality: testimony, [indirect] evidence (e.g., chocolate eggs, or money left under a pillow), and *'engagement in rituals'* (e.g., leaving cookies out for Santa, hiding teeth, or actively searching for chocolate eggs; p. 1502) Additionally, Goldstein and Woolley [14] claim that [direct] evidence (i.e., engagement with Santa at a mall) reduces skepticism and predicts belief. Children use multiple forms of testimony to build beliefs about real and non-real figures, allowing for fantasy-reality distinctions that extends beyond a simple binary evaluation of real or not-real, and which may allow for sophisticated sub-categories and hierarchies (for a review see: [15,16].

Cultural rituals and norms are only infrequently discussed as a form of testimony in the literature (perhaps due to the difficulty of experimental manipulation). And yet, there exist evidence of its importance. Woolley, Boerger, & Markman (2004) created a fictional agent called 'The Candy Witch' in order to examine how a child's understanding of supernatural figures develops. They created an induction process that involved two different experimenters arriving at a school on two separate occasions in the week preceding Halloween during which they described and presented an image of the Candy Witch (in this manner they created a sense of consensus). Children were told that The Candy Witch is a friendly witch who trades toys for the candy that children collect at Halloween. In order to summon the Candy Witch, there were multiple behavioral requirements: the child needed to refrain from eating some proportion of their collected candy to pay the Candy Witch, while the child's parent needed to phone the witch to arrange the transaction. Across experiments this induction method led a majority of children to report that the Candy Witch was real [17,18]. Among older children, participation in the behavioral aspects significantly increased endorsement of the reality of the Candy Witch above those who only heard about the witch, with some weak evidence suggesting that those who were visited, compared to those who were not, recalled more details about the Candy Witch one year later [17,18].

In contrast, Piazza, Bering, and Ingram [19] created Princess Alice, but were unable to arouse in children a response consistent with belief in her. Children were simply provided testimony of Princess Alice: that she was magical, invisible, and present in the room. Children later engaged in a task in which they were afforded the opportunity to cheat. While 11 children (48%) professed belief in Princess Alice (5 were unsure, 7 did not) the 'presence' of Princess Alice did not change their likelihood of cheating. Indeed, belief in Princess Alice did not significantly correlate with the child's attempt to explore the place where she was ostensibly located

in the room, nor even with looking in her direction (subsequent research has also failed to find positive results; see Kapitány, Reindl, Nielsen, under review; [20]. These results suggest that children's beliefs are not so easily manipulated. We describe Princess Alice and the Candy Witch as generalizable for larger real world beliefs, and argue that the degree of behavioral involvement required on the part of children and the conspirators aids belief. Thus, children may view the existence of some agents as being more plausible than others, and the key to this distinction may be the associated behavioral requirements.

Behavior stipulations for figures may also be expressed as cultural and community rituals. Henrich (2009) has argued that the 'costliness' of certain cultural actions are linked to the evidential value of the act—one would not perform such strange actions (like erecting a dead tree indoors and decorating it) if they did not think such actions were justified by sincere belief. Broadly, the actions associated with beliefs, groups, and specific figures are culture-specific rituals [21–25], which are argued to be necessary for the transmission of culture-specific values and beliefs [26]. Consider Christmas: many families independently perform the same kinds of behaviors for Santa, such as putting out milk and cookies (and a carrot for the flying reindeer), erecting and decorating a tree inside a house, wrapping presents, and eating certain kinds of food [27] in whole (or part) *for* Santa Claus. These behaviors are not just consistent within families over time, but consistent between families within culture. Given that diverse consensus of testimony leads to belief [18], diverse consensus of *action* may be just as (and possibly even *more*) important. Rituals provide proxy evidence for the existence of such characters upon which children rely, as they demonstrate the actor's dedication to, or conviction for, the specific figure [28]. Thus we argue that figures who require specific behavior under certain conditions, which includes the performance of cultural rituals and norms, are more easily accepted and endorsed as 'real' by children than figures who do not—particularly at the aggregate level.

This study is the first attempt to present children with a range of real, cultural, and fantastical figures in order to determine whether children build a hierarchy of endorsement. We anticipate that children will vary in the degree to which they endorse these figures as real, consistent with a) the degree of direct evidence, b) the degree of indirect evidence, c) whether or not a child performs certain actions or rituals for the figure, and d) whether a figure is endorsed explicitly as fictional. We contend that certain kinds of rituals performed under specific conditions—performed to appease or summon a figure—are a powerful method for strengthening a child's endorsement for a figure. For example, erecting a tree inside the house, or hunting for eggs, only happens on a specific period of the year; and only teeth (and not toenails) are placed under a pillow when they are lost from the body. We also anticipate that, based on the four qualities outlined, that a hierarchy of endorsement will be apparent in children's belief scores —such that relative differences between figures will be apparent and 'sensible'.

Of note, we use the term *"figure"* to broadly refer to all agents, entities, or persons presented in our study (irrespective of other qualities); and the term 'sensible' to describe the endorsement of these figures by adults. Finally, it is important to consider what exactly children and adults understand the term 'real' to mean. While we grapple further with the nature of this question in the discussion, it is worth noting that we approach this problem from two angles. We conceptualize responses to the question '*Do you think [the figure] is real*?' as 'endorsement' [of reality status], rather than of belief [in reality status] *per se* (since we did not measure any behavior that might provide stronger evidence of a true conviction in a figure's reality). That is, 'belief' describes the degree to which a participant authentically holds that a figure is real, while 'endorsement' can only be regarded as a self-report measure. We would also stress that this is a study focussed on the cultural processes that sustain endorsement at the level of

populations, and not a specific study in individual differences or longitudinal developmental trajectories.

Here we present 'The Pantheon', a list of figures and categories, as well as the associated beliefs and rituals for each figure that are typical in western contexts. In Table 1 we define relevant terms associated with our figures, and in Table 2 we outline the figures and their expected groupings within the pantheon. We anticipated the existence of these categories with respect to the 1) available direct evidence for the figure, 2) indirect evidence for the figure, 3) behavioral rituals and cultural norms associated with the figure, and 4) presence or absence of an explicit 'fiction' status. The categories and figures are presented in order of expected endorsement, as a function of these four features.

## The figures

**Real figures.** Our study includes two Real Figures (aggregated into the 'Real' category)—an adult known to the child, but who is not particularly close to the child (selected by the parent, and to be someone like a family friend, a doctor, or teacher) and The Wiggles. The Wiggles are an Australian band who perform children's music on a long-running TV show, and who regularly tour Australia, The US, The UK, and the United Arab Emirates. The Wiggles (or at least, the constituent members) are real humans, dressed like humans, and who appear on television as themselves—they are non-animated, and have typical physical and biological limitations. However, The Wiggles also share many qualities associated with animated, absurd, and impossible fictional agents, such as appearing on television, singing and dancing, and having friends who are impossible and who go on fantastic adventures. Given The Wiggle's extensive touring, children may or may not have had the opportunity to directly interact with these figures—they are, however, extremely popular and widely known in Australia (the population under consideration). We expect by virtue of these figures' obvious correspondence to human models in the child's life that children understand that interactions with the Wiggles are in principle real (just as they are with real but non-present individuals in their lives), and that any interactions with The Wiggles will be governed by the same behavioral expectations and norms as any other real person, that they will report that The Wiggles are (highly) real, like a truly real person in their lives. Though it is the case that Wiggles are primarily known to the

**Table 1. Definition of terms.**

| Term | Definition |
| --- | --- |
| 'Direct Evidence' | One has direct evidence of a thing when one *has directly* interacted, or in principle *can directly* interact, with the thing in question, via ordinary means.*e., Charlie has direct evidence that the Eiffel Tower exists, as they have climbed it.* |
| 'Indirect Evidence' | One has indirect evidence of a thing when one may not directly interact with a thing, but can interact with a proxy of a thing. Direct interaction with the thing would require extraordinary means. *E.g., When Sammy thinks of a Homo neanderthalensis, they think of a walking, talking, cultural being who lived as recently as 40,000 years ago. Sammy has seen the bones of Neanderthals (and, as such, has indirectly interacted with them), but has not directly interacted with walking, talking, cultural being. It would be impossible to directly interact with a Neanderthal.* |
| Cultural rituals and norms | 'Ritual' for short; A term used to denote a set of behavioral norms and requirements performed in association with a thing. *E.g., It is Peter's birthday, they are turning 8. Peter expects a sweet cake with 8 candles in it, people to sing 'happy birthday', and to receive gifts. Peters parents and guests anticipate satisfying these expectations.* |
| Fiction | A thing that is not veridical, and known to have been the product of an intentional act of creation. Fictional things do not often make claims to being real. *I.e., Harry Potter, while ostensibly human, is fictional, and is regarded as an intentional product of the mind of J. K. Rowling.* |

**Table 2. The categories of figures.**

| Category | Qualities / Criteria | Examples included |
|---|---|---|
| Human/Real Figures | Figures that are both human and extant. They are real and are presented as such. Children have direct evidence, or recognize that direct evidence/interaction is possible (by virtue of the figure's humanness). As with all humans, there are norms associated with the figure. | A person known to the child, The Wiggles |
| Cultural Figures | Figures that do not exist, but are presented to the child as real, and done so culturally. These figures have cultural and social norms associated with their cosmology, such as rituals, and children receive indirect evidence in the form of gifts (ostensibly from the figure in question). | Santa, The Easter Bunny, Tooth Fairy |
| Ambiguous Figures | Figures that were real, or are possibly real, and are presented to the child as such. Children have received, or may have received, indirect evidence of these figures' existence (as with dinosaur fossils, and other kinds of representations in various media). | Aliens, Dinosaurs |
| Mythical Figures | Figures that are not real, which no standard norms associated with their endorsement at a cultural level, but which parents may idiosyncratically endorse. Children typically do not receive direct- or indirect evidence for these figures' existence. | Unicorns, Dragons, Ghosts |
| Fictional Figures | Figures that are not real, and are presented as works of fiction, which a child is likely to have had exposure to on television or other media. | SpongeBob, Peter Pan, Elsa |

child via medium of TV, we believe this makes them an appropriate comparison group to the other figures (who are also known primarily via same and similar mediums).

**Cultural figures.** Cultural figures share many similarities with real figures, despite the fact that they are not, in fact, real. Much like interacting with a human, cultural figures are associated with specific norms and behavioral stipulation. Additionally, from the point of view of the child, cultural figures also provide indirect evidence of their existence in the form of gifts. Moreover, they are broadly endorsed as real by reliable models in the child's life, and in the media. However, the child does not receive direct evidence of the figure.

One notable exception to this statement is Santa. In many parts of the Western world, a typical Christmas time ritual for parents is to take their children to visit a live actor dressed as Santa in a public place, to tell children that this is the real Santa, to take a picture of the child with Santa, and for the child to tell Santa of their desired gifts [8]. Parents cite as their reason for visiting a mall santa their desire to make-believe and play with their children, as well as to increase their children's belief in Santa [14]. Though we contend that meeting a magical simulacrum in a mall or shopping is not quite the same as seeing, for example, a family doctor. It is an empirical question whether or not this makes a difference in endorsement for Santa relative to the Easter Bunny or the Tooth Fairy—and it appears that endorsement between these figures is not statistically different (as revealed by our own data).

**Ambiguous figures.** While real figures and cultural figures share many similarities the relationship of these two groups to ambiguous figures is of particular interest. By ambiguous, we mean that children are exposed to ideas about these figures, mostly presented either as real (or at least possible), but without ritual or behavioral requirements, nor direct evidence of existence. Dinosaurs and aliens may exist in the world, or may not—but their existence does not impact how the child lives their lives. When children are exposed to these 'ambiguous' figures they are forced to make difficult evaluations. Frequently such figures are presented as real in principle, despite the fact that they share many qualities with figures that are real and not real respectively. Dinosaurs are similar to both lizards (real) and dragons (not real); while aliens are similar to any novel biological entity (say, a Capybara) even though the child may never

have seen or experienced one first hand. While not perfectly equivalent, in the context of the four factors of testimony (direct- and indirect evidence, rituals/norms, and fictional disclaimers) that lead to endorsement, dinosaurs and aliens seem approximately equivalent: Ambiguous figures are endorsed as real by reputable individuals and may be supported by indirect evidence (various kinds of physical representations), but–importantly—are bereft of cultural rituals or specific requirements for action. (We note that Aliens may not have indirect evidence for their existence, per se. However, to the extent that they are biologically plausible and may have vehicles that can travel into space (as we humans do), we have somewhat liberally defined them as ambiguous. Had we not, we may have had particular trouble in accounting for an adult understanding of these figures). A 'reasonable' understanding of such an agent would be to admit a degree of ambiguity, without dismissing the possibility of their existence (or non-existence) entirely.

**Mythical figures.**   Mythical figures are not real, provide no direct or indirect evidence of themselves, and are not associated with any kind of culturally prescriptive actions, but which parents may, or may not, idiosyncratically endorse as real via testimony (e.g. Unicorn; [29]). While we recognize that myth is culturally dependent, the distinction between a mythical and a cultural figure in this conceptualization is that cultural figures are associated with cultural rituals. When reading a story, for example, unicorns, dragons, and ghosts may be presented as real within the context of the story, and may even generate strong emotions in children, but their presentation is not culturally standardized; dragons may be cruel (per The Hobbit) or brave and kind (How to Train Your Dragon), unicorns may be pure (per Harry Potter) or murderous (Cabin in the Woods), while ghosts can be friendly (Casper), vengeful (Hamlet), lecherous (Beetlejuice), or paternalistic (Star Wars). Importantly, none of our proposed mythical figures are associated with a standard corpus of behavior that supports their belief. Note also that we are discussing a generic mythical agent (i.e., a 'dragon' rather than 'Smaug' or 'Toothless'), and that such generic agents may receive occasional testimony in support of their existence (which is likely different from specific instantiations).

**Fictional figures.**   Finally, in contrast to real, generic mythical, and cultural figures are 'fictional figures', such as Peter Pan, Spongebob Squarepants, or Princess Elsa. These figures are not endorsed as real by adults, rather, they are explicitly presented to children as fictional; they are not associated with forms of evidence in favour of their existence, and are not associated with any cultural practices. In these instances, these figures ought to be *prima facie* not-real, and should occupy the tail end of a hierarchical scale of reality distinctions.

## Study 1. Online data collection

We examined endorsement for a range of figures among children (aged 2–11), and adults (over 18 years) in order to test the hypothesis that such figures feature in a sensible and relative hierarchy of endorsement (where 'sensible' is defined relative to adult's endorsement). We did so using a parent-as-researcher model, which we validated using more traditional methods in study 2. Parents were recruited via online advertisement and mailing lists, and gave consent via survey check-box.

The current study expands on previous work by comparing levels of endorsement in different kinds of figures [3,30], with specific reference to the four qualities associated with figures/categories: direct- and indirect-evidence for reality status, ritual and normative behaviors, and endorsement of fictional status. In so doing, we hope to empirically describe how children are evaluating various kinds of cultural representations relative to each other, and whether we see evidence for ambiguity within the spectrum of endorsement.

The figures we presented to children were chosen based on whether their exemplar characteristics allowed them to fit clearly in one of our theorised categories (see Table 2). Both adult and child participants' provided an endorsement score for the realness of each figure on a 9 point likert scale, as well as responding to questions related to the epistemological qualities of a subset of figures (see S1 Data)

## Stimuli, hypotheses, and analyses plan

We made the following predictions regarding our data:

**H1:** We will observe the expected categorization of figures based on endorsement scores.

**H2:** Children's endorsement will conform to a hierarchical pattern consistent with the degree to which they are culturally and evidentially supported.

**H3:** Adults' endorsement of the realness of figures will be high for 'real figures', lower for ambiguous figures, and near floor for cultural, mythical, and fictional figures.

**H4:** As age increases, children's endorsement patterns will become more adult-like.

## Methods

### A note on data collection and analyses

Data collection for this experiment was executed over the course of 12 months in a longitudinal manner, and several additional hypotheses were proposed for this larger within-participants dataset. However, due to attrition we were not able to perform these planned analyses. We have included descriptive, but not inferential, statistics of our longitudinal data in S1 Data, as well as other information about this broader research effort. What follows is a description and analysis of data from the first wave of data collection. We replicated our findings and present them in study 2).

### Recruitment procedures and participants

Using our lab databases, as well as our facebook page, we invited Australian parents to enroll their children in a year long study examining their child's endorsement for various kinds of figures. The first wave of the survey was conducted in July 2015 over a 3 week period. The administration of the study involved an online survey that the parent conducted with their child/children. In order to avoid incomplete responses and to avoid causing offence to parents, we asked parents to indicate which of the figures they were comfortable asking their children about. This was done to ensure that parents could omit any figures they felt were inappropriate (for example, some parents may wish to preserve in their children a belief in Santa, and so they were able to opt-out of Santa-related questions. This explains the varying number of responses between figures). In addition, parents agreed to a set of experimental protocols (common to any lab based study) to minimize bias (see S1 Appendix).

A total of 154 children provided responses during the first wave of data collection. Surveys that did not include an answer to the final question of the survey were excluded (which was the only question asked of the parents specifically: *Do you think your child was being truthful with you*?), among those that failed to answer the final question, none responded to more than 3 figures. A total of 95 children were included in the final analysis.

An adult comparison group (N = 57) was recruited from an undergraduate research pool in exchange for course credit. Adults were asked to participate in this study after completing an unrelated study on facial expressions.

## Procedure

**Training questions.** Children first participated in a number of practice questions in order to become familiar with the 9-point likert scale, which was visually represented as 9 grey stars which turned gold when clicked. These questions related to obviously false things, obviously true things, and confusing things. The goal of this task was to get children using the whole scale (see S2 Data for the full list).

Children were asked whether 'Elephants' and 'Chairs' were real, and data suggested the question was understood appropriately (respectively, M = 8.51, SD = 1.63; M = 8.4, SD = 1.99). On the more challenging questions of whether 'Floating Rocks' (M = 4.64, SD = 3.41; 'Short-nosed Elephants' (M = 4.07, SD = 3.36) were real, it was clear, that the mid-points of the scale were used and understood by the sample, and that skewness and kurtosis were normal. The lowest scores were associated with the obviously false things, such as 'singing chairs' (M = 3.30, SD = 3.20); and 'Upside down trees' (M = 3.38, SD = 3.27).

**Target questions.** Children were asked to rate the realness of the 13 target figures (presented in random order) on the 9 star likert scale, by answering the question "*Do you think [figure] is real*?", with 1 stars indicating "*not at all real*" and nine stars indicating "*definitely real*". We reduced all values by 1, so that '1' responses—which qualitatively represented 'not at all real' were now represented by '0'—which we argue is a more interpretable value. During the first wave of data collection we failed to collect data on a real person (who is known to the child). This was an oversight. Data on this target were collected at T2 and have been used during these analyses where appropriate.

## Results

### Descriptive statistics

A total of 95 children were included in the final analysis. The mean age of these children was 5.12 years (SD = 2.17); the youngest child was 1.97 years, while the oldest was 11.10 years (age was calculated by determining days passed since the child's reported birthday and the day of testing). (As described previously, parents read the questions to their children, and honestly reported their child's responses. While some may argue that a 2-year old could not provide useful data, we believe this is data worth having. We note that if we remove the three 2-year olds that results for the analyses presented hereafter do not vary. Our full data are available at https://osf.io/wurxy/, and we invite readers to download it). Sex data was not collected. A total of 57 adults constituted the adult sample. Their mean age was 20.23 years (SD = 5.31). Eighteen of these participants were raised overseas, and all spoke English at a tertiary/university level. A single adult, aged 48, was 5.23 standard deviations above the mean for age, and appeared to influence several regression values—inasmuch as we hope to make age-relevant claims, this outlier is unduly influencing the predictive value of age as a predictor. For this reason, this single individual was removed from regression analyses.

### Hypothesis 1: Do figures form into categories?

Our first hypothesis was that the figures would fall into five groups—real, cultural, ambiguous, mythical, and fictional (see Table 2)—due to the specific qualities associated with figures described previously. We tested this hypothesis using a cluster analysis [31]. Here, we have a

population of children, who each provided a reality endorsement of up to 12 figures (or 12 'variables'). Our aim is to identify whether those figures (variables) cluster together into 'homogenous and distinct' groups [32,33]. Note: At T1 we did not collect a child's endorsement of a real person known to them (though we did at all subsequent time-points). Thus, we did not include 'real person' in the cluster analysis.

The method of clustering employed here is the hierarchical method (using *iclust* function of the R-package *Psych;* Revelle, 2016). First, we produce a correlation matrix (see Table 3) of the variables to cluster. If we first accept that each participant has provided one observation per figure (in this case, there are 12 observations as there are 12 figures), then we can conceive of each of these observations representing their own cluster, thus, there are 12 clusters. The next step is to aggregate the two most highly correlated variables (figures) into a cluster, producing a total of 11 clusters. For each repetition of this aggregating procedure, there is a corresponding decrease in number of clusters. The *psych* package repeats this process until one of two measures of internal consistency fail to increase (cronbach's alpha, α, or worst-split-half-reliability, β; respectively; [31,34], or until there is only one cluster. An assumption of cluster analysis is that the variables are not too strongly correlated with each other variable (i.e., $r > .9$; [32,33]. Table 3 shows the correlation matrix of endorsement scores for each figure (note: only one of 12 correlations between figures exceeds this threshold—between Peter Pan and Spongebob; $r = .945$ and so does not pose substantive issues in interpretation). Meanwhile, the heuristic-rule for sufficient power to extract factors can be described by the formula: $2^c$ (where 'c' is number of expected clusters, which in this case is five; [35]. Thus, we have sufficient power to detect the 5 clusters predicted. In our case, our analysis created one super-ordinate cluster, which is to say, that the lower-order clusters have the same measures of reliability independent of whether we accept the single-cluster super-ordinate outcome, or a greater number of small clusters. Because our hypothesis for the clusters were clear, and because the reliability values remain the same at all levels of aggregation, we extracted 5-clusters (see Fig 1).

The clusters generated and presented in Fig 1 are fairly interpretable, however, it is not necessarily clear why 'Fictional (A)' and 'Fictional (B)' are distinct. Allowing for a four-cluster solution (see Fig 2), we find that Fictional A and B cluster together. Since the values of clusters within the hierarchy do not change with level of analysis, we retain a 4-cluster solution for the sake of interpretability.

Consistent with our hypothesis, 'Cultural Figures' clustered reliably. We expected that Dinosaurs and Aliens would cluster together, however, we found that Aliens clustered with Dragons and Ghosts (the latter two were expected to cluster). We have termed this the *ambiguous figures* cluster, and we explore the nature of this cluster in the discussion. Similarly, we note that the *mythical* category has dissolved, such that unicorns now cluster with *fictional* figures. We have no immediate interpretation for this, except to say that the initial distinction between fictional and mythical may have been too fine. The primary notable difference is the new *'virtually real'* category: Dinosaurs clustered with The Wiggles. We believe this is because both are real, and both figures only provide indirect testimony for their own existence. We interpret this as attributable to other forms of testimony children receive regarding the reality status of these figures; we note also the relatively low reliability statistics (which are a consequence of moderate correlation even if absolute endorsement appear to vary; i.e., Wiggles are endorsed more highly than dinosaurs in aggregate; see Fig 3).

## Hypothesis 2: Do figures (and their categories) form a hierarchy?

We predicted with Hypothesis 2 that children's endorsement will conform to a hierarchical pattern consistent with the degree to which agents are culturally and evidentially supported.

**Table 3. Correlation matrix (r values) of 'realness' scores for figures.**

|  | Wiggles | Santa | Tooth Fairy | Easter Bunny | Aliens | Dinosaurs | Ghosts | Unicorns | Dragons | Spongebob | Princess Elsa | Peter Pan |
|---|---|---|---|---|---|---|---|---|---|---|---|---|
| **Wiggles** | - | | | | | | | | | | | |
| **Santa** | 0.065 | - | | | | | | | | | | |
| **Tooth Fairy** | -0.085 | .548** | - | | | | | | | | | |
| **Easter Bunny** | 0.072 | .741** | .697** | - | | | | | | | | |
| **Aliens** | -0.19 | .514** | .478** | .481** | - | | | | | | | |
| **Dinosaurs** | .516** | 0.071 | 0.031 | .276* | 0.175 | - | | | | | | |
| **Ghosts** | 0.17 | .356* | 0.281 | .364* | .514** | .321* | - | | | | | |
| **Unicorns** | 0.01 | .547** | .486** | .573** | .593** | 0.289 | .416* | - | | | | |
| **Dragons** | 0.047 | 0.257 | .428** | .395** | .583** | .308* | .453** | .668** | - | | | |
| **Spongebob** | 0.281 | .416* | 0.269 | 0.348 | 0.337 | 0.174 | 0.294 | .701** | .518** | - | | |
| **Princess Elsa** | 0.138 | .431** | 0.282 | .448** | 0.323 | 0.205 | 0.143 | .854** | .386* | .739** | - | |
| **Peter Pan** | 0.195 | 0.413 | 0.542 | .463** | 0.387 | 0.203 | 0.385 | .848** | .422* | .945** | .793** | - |

* deontes p < .05

** denotes p < .01

Fig 3 shows the descriptive statistics (Mean and 95%CI) for children's endorsement of each figure and each (revealed) category, respectively.

In order to determine whether there was a sensible hierarchy of endorsement, based on the derived categories, we conducted a Multilevel Model in which age was the predictor variable and 'belief' was the outcome variable, where we allowed y-intercepts to vary (i.e., we let each category of figures have it's own y-intercept value) and freely estimated slopes (i.e., the beta value associated with age could vary as a function of category). Note that the results were conducted on the categories of figures and not the individual figures. We imputed missing values using the *mice()* package [36], which imputes missing values for each variable based with separate regression models (with stochastic variation), each of which includes values from all other variables. For example, a missing value for 'Fictional Figures' is imputed based on the beta value of age, as well as the beta values associated with each other figure category. The stochastic

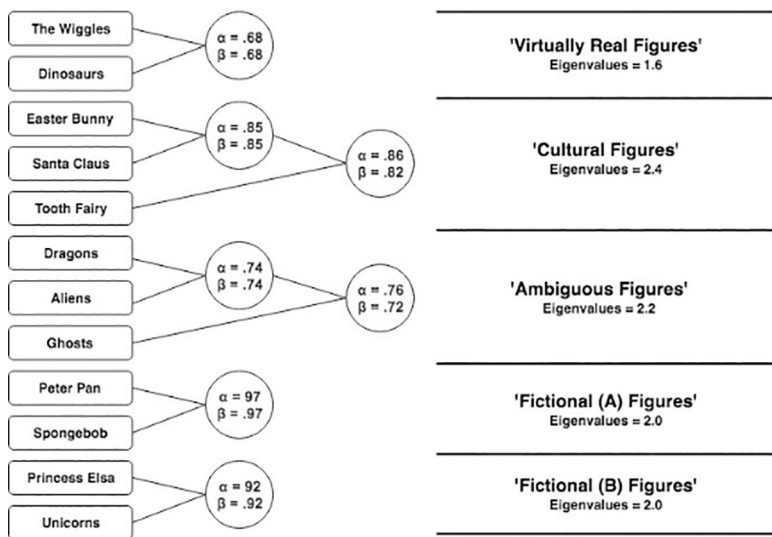

**Fig 1. A 5-cluster solution of the child's pantheon.**

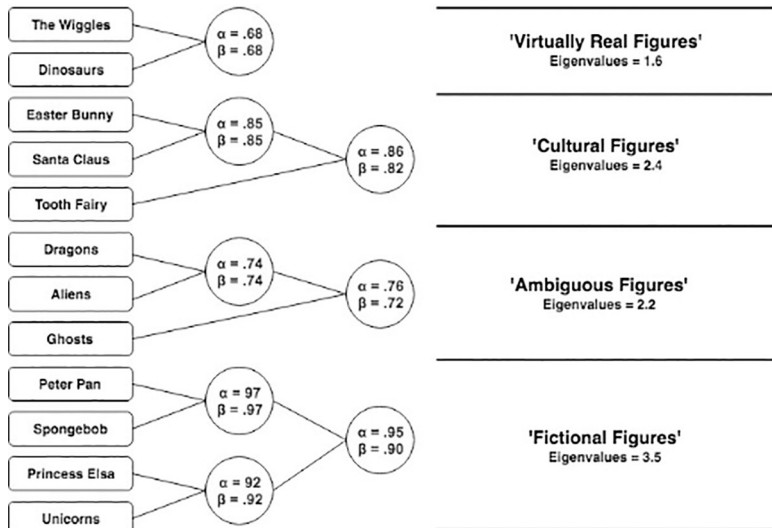

**Fig 2. A 4-cluster solution of the child's pantheon.**

variation includes an estimated error term, in so doing, each datapoint is imputed with variance around the regression line—the value of this is that it compensates for an overestimation associated with regression-imputation alone [36].

We identified that a model with a fixed y-intercept (Model I) for all categories was significantly improved by a model with freely estimated y-intercepts (Model II), SD = 1.27, $\chi^2(3)$ = 103.14, $p < .001$, suggesting a multilevel model might be appropriate. Next, we introduced age as a predictor (Model III); Model III was a significant improvement on Model II, $\chi^2(4)$ = 10.94, $p = .001$, indicating age (as a fixed effect for all categories) predicted endorsement. The correlation between the category codes and regression intercepts $r = -.411$, indicating that as age increased endorsement decreased. Next, we introduced freely estimated slopes for age (Model IV) as we expected that age would be differentially predictive of endorsement according to category. We found that Model IV was a significantly better fit than Model III, SD = .29, $\chi^2(6) = 32.62$, $p < .001$, and the slopes and intercepts were correlated, $r = -.628$. Fig 4 shows the results of Model IV (including β values), and Table 4 shows regression values.

It is important to note that the beta values associated with any given category (per Fig 4) should not be interpreted independently from the other beta values within the model. That is, Fig 4 and Model IV reveal that age interacts with category to predict endorsement.

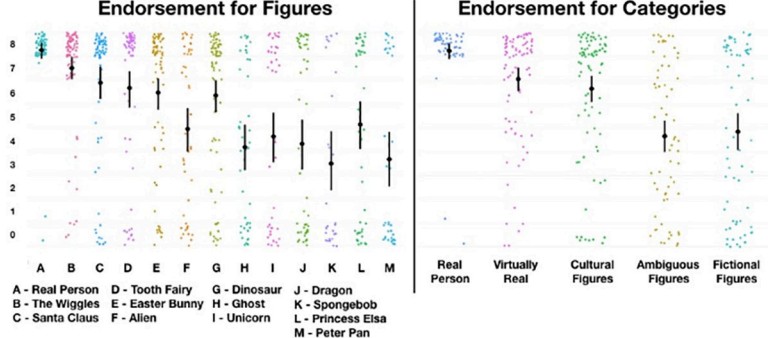

**Fig 3. Childrens endorsement scores for individual figures and categories.**

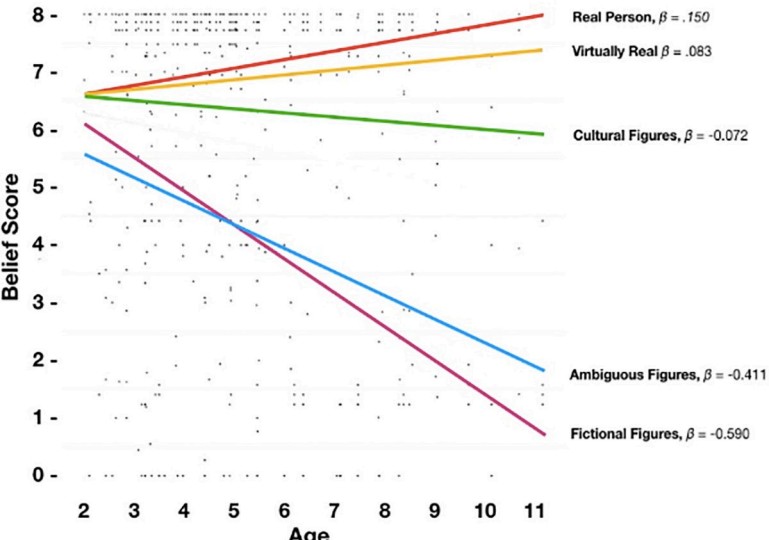

**Fig 4. Results of the multilevel model in which a child's age interacts with category of figure.**

### Hypothesis 3: What does an 'adult-like' understanding look like?

Our third hypothesis was that adults' endorsement of the realness of figures would be high for 'real figures', lower for ambiguous figures, and near floor for cultural, mythical, and fictional figures.

Fig 5 shows adults endorsement of all individual figures. As can be seen, endorsement for the majority of figures is effectively zero, with very little variation. Due to this low variation we cannot conduct the cluster analysis to determine equivalent categories. However, visual inspection reveals that adults endorse real figures near ceiling, and the figures constituting 'cultural, mythical, and fictional' categories at floor. The only notable exception is that of Ghosts, which are endorsed with a mean value of 2.26 (SD = 2.89), suggesting considerable idiosyncratic variation. Aliens are endorsed at middling rates, while dinosaurs appear to be endorsed at high rates by most participants, and floor rates by a few. Thus, we suggest there are three categories for adults—real, unreal, and ambiguous (with ghosts and aliens constituting the latter).

### Hypothesis 4: Does age predict a more adult-like understanding of figures?

We predicted that as children age their endorsement will become more adult-like. It is clear from the data presented in Fig 4 that the negative slopes associated with fictional characters that age negatively predicts endorsement between participants. That is, endorsement for figures within that categories declines with age—which, predictably, will converge on values of

**Table 4. Results of iterative regression models for determining hierarchy of belief.**

|  | Model I | Model II | Model III | Model IV[x] |
|---|---|---|---|---|
| **Age (Beta; SE)** | - | - | -.168 (0.051)*** | -.168 (0.142) |
| **Constant (int; SE)** | 5.774 (0.124)*** | 5.774 (0.579)*** | 6.643 (0.636)*** | 6.643 (0.357)*** |
| **Observations** | 475 | 475 | 475 | 475 |
| **Log Likelihood** | -1144.203 | -1092.631 | -1087.159 | -1070.850 |

*** p< .01; x = Final Model.

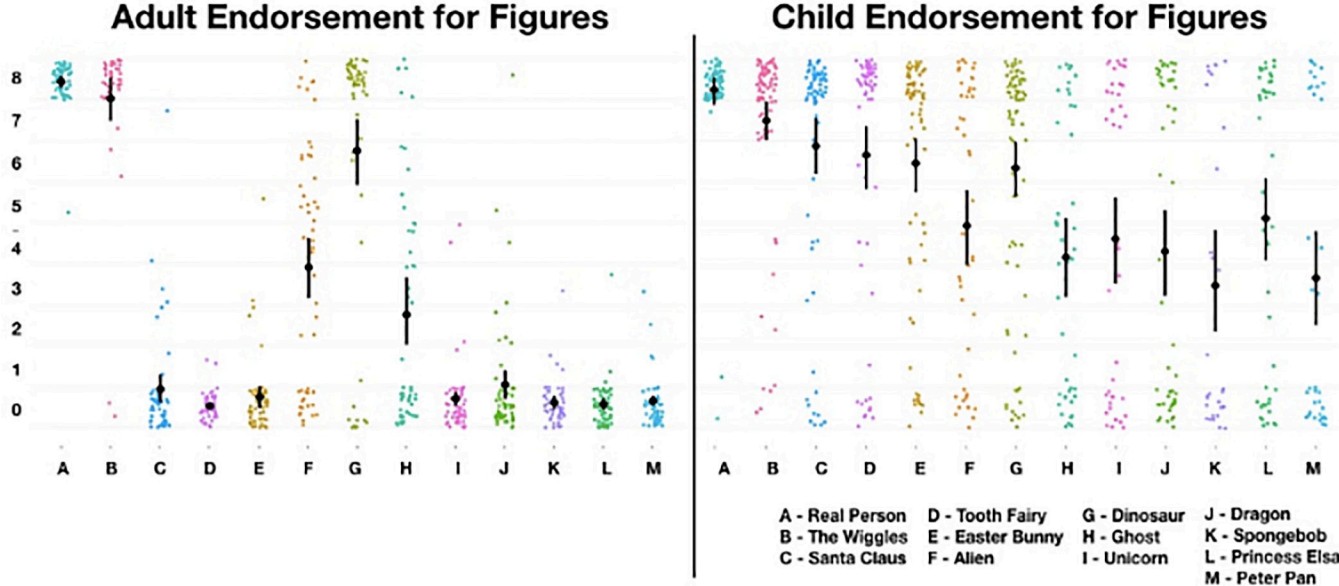

**Fig 5. Endorsement for figures for adults and children.** Note: Child Figure is a reprint of Fig 3.

endorsement reported by adults for the same figures. With regard to real and 'virtually' real individuals, the positive slope suggests increasingly adult-like endorsement—that is, as children age, they recognize the reality of those figures (or perhaps, are better able to sensibly interpret the question). Notably, the slope for Cultural Figures, while nominally negative, is largely flat. We have two possible explanations for this: the first is that children who do not believe in these cultural figures, when reporting to their parents, are partially motivated to deceive their parents that they actually do. The second is that belief in cultural figures is likely an S-shaped curve, where belief declines precipitously around the ages of 7 or 8 [37], and that having fit a linear model, we are unable to capture the true nature of this effect. Study 2 addresses the former concern. We also note that our data contains proportionally fewer children over the age of 7 than it does under. Unfortunately, only future research will address this.

We observe three intermediate values among adults, aliens, dinosaurs, and ghosts (see Fig 5 for all values) among the 'ambiguous figures'. By the spread of data, aliens and ghosts are regarded as ambiguous as evinced through the use of middling values; dinosaurs are generally strongly endorsed through a skew is introduced by a minority who do not affirm their reality. We note that the difference between adult and children scores was not significant for Aliens, t(91.89) = 1.616, $p$ = .110, and Dinosaurs, t(109.95) = .809, $p$ = .421 (as were non-parametric tests). However, adults collectively endorsed Ghosts at significantly lower levels than did children, t(89.073) = 2.246, $p$ = .027.

## Discussion

In study 1 we observed that a cluster analysis broadly supported our prediction that figures would cluster into specific categories. Importantly, cultural figures—those associated with cultural rituals and specific behaviors—formed their own category, while other figures formed clusters with similar-others, even though some specific elements were off. We observed, through the use of a multilevel model, support for the claim that these different categories had differential base-rates of endorsements, consistent with the proposed hierarchy (as evinced by

the model fit in which y-intercepts varied), and that belief was predicted by age differentially according to category. Relative to adult scores, children's scores revealed that developmental changes would produce broad convergence on adult-like (or 'sensible') levels of belief.

We note that some degree of imputation was used to arrive at these results, and that some aspects of data collection (such as the use of parents-as-research-assistants) may raise questions about the accuracy of our findings. Thus, we conducted study 2 using more traditional methods of data collection in order to address these limitations.

## Study 2. Research assistant collected data

**Recruitment procedures and participants.** Between the months of April and July, a trained research assistant (RA) from [redacted university] attended a Science Museum [details redacted] in a capital city of Australia, in order to collect a corresponding sample of children. The protocol was the same as with study 1, save for the following changes. The RA approached parents in the foyer of the public science museum, and invited them to involve their children in this research project. Parents were then briefed, and gave consent for their child to participate. Parents indicated which figures the child *should not* be asked about by the RA, and provided the name of a real person known to the child. Data was collected using an iPad, and children were encouraged to interact with the iPad themselves.

Due to operational and time constraints, we were able to collect responses from 82 children (though one child was excluded for not completing the task). In total, our final dataset included 81 children. The mean age of these children was 6.17 year (SD = 1.79) [Original mean = 5.12, SD = 2.17]. The youngest child was 3.44 years [original 1.97 years] while the oldest was 11.67 [original 11.10 years]. As can be seen in Table 5, rates of consent for each figure were considerably higher in the replication dataset, as were the proportion of responses to permitted figures.

## Procedure

**Training questions.** Children first participated in a number of practice questions in order to become familiar with the 10-point likert scale, which was visually represented as 10 stars which turned gold when clicked. These questions related to obviously false things, obviously true things, and confusing things. The goal of this task was to get children using the whole scale (see S1 Data for the full list).

**Table 5. The proportion of children for whom permission was granted for each figure (in each dataset), and the proportion thereof who subsequently provided endorsement scores.**

|  | Original Dataset | | Replication Dataset | |
|---|---|---|---|---|
|  | **Permission** | **Response** | **Permission** | **Response** |
| Real Person | - | - | 1.00 | 0.90 |
| Wiggles | 1.00 | 0.85 | 1.00 | 0.86 |
| Santa | 0.94 | 0.78 | 0.99 | 0.91 |
| Tooth Fairy | 0.93 | 0.69 | 0.99 | 0.93 |
| Easter Bunny | 0.94 | 0.81 | 0.99 | 0.89 |
| Alien | 0.91 | 0.55 | 0.98 | 0.95 |
| Dinosaur | 0.98 | 0.85 | 1.00 | 0.88 |
| Ghosts | 0.88 | 0.54 | 0.94 | 0.84 |
| Unicorns | 0.94 | 0.51 | 0.99 | 0.90 |
| Dragons | 0.96 | 0.54 | 0.96 | 0.87 |
| Spongebob | 0.88 | 0.36 | 0.99 | 0.83 |
| Princess Elsa | 0.95 | 0.53 | 0.99 | 0.85 |
| Peter Pan | 0.91 | 0.41 | 0.99 | 0.83 |

**Target questions.** Children were asked to rate the realness of the 13 target figures (presented in random order) on the 10 star likert scale, by answering the question "*Do you think [figure] is real?*", with 1 stars indicating "*not at all real*" and 10 stars indicating "*definitely real*".

## Results

All analyses were identical to those of Study 1, and the R code we used to conduct the study is available on open science framework (https://osf.io/wurxy/)

### Hypothesis 1: Do figures form into categories?

Table 6 documents the correlation matrix. No correlation exceeded .9, and so the pattern of relations does not substantially challenge our analysis; Per our original analyses, we ran a cluster analysis on our data allowing for 5 categories, and not including 'Real People' (see Fig 6). As before, we found that a 'Cultural Figures' cluster emerged, as well as a 'Fictional Figures' cluster. However, we observed some differences: Dinosaurs and The Wiggles did not aggregate (in the primary dataset this cluster had relatively low reliability). Dragons, Aliens, and Ghosts aggregated as before, but this time the cluster also included Unicorns (which had previously aggregated into the fictional figure category).

### Hypothesis 2: Do figures (and their categories) form a hierarchy?

We predicted that children's endorsements will conform to a hierarchical pattern consistent with the degree to which agents are culturally and evidentially supported. Below, we show descriptive statistics (Mean and 95%CI) for children's endorsement of each figure and each category (Fig 7). Note that for Fig 7 that the original data is plotted 0–8, and replication data is plotted 1–10. We are not making a direct inferential comparison between the two dataset (as

**Table 6. Table of correlation between figures (with correlations from main manuscript in parentheses).**

| | | 1 | 2 | 3 | 4 | 5 | 6 | 7 | 8 | 9 | 10 | 11 | 12 | 13 |
|---|---|---|---|---|---|---|---|---|---|---|---|---|---|---|
| 1 | Real Person | - | | | | | | | | | | | | |
| 2 | Wiggles | -0.05 | - | | | | | | | | | | | |
| 3 | Santa | -0.07 | -0.07 (0.07) | - | | | | | | | | | | |
| 4 | Tooth Fairy | -0.04 | -.13 (-0.09) | 0.83 (0.55) | - | | | | | | | | | |
| 5 | Easter Bunny | -0.07 | -.09 (0.07) | 0.86 (.74) | 0.83 (.70) | - | | | | | | | | |
| 6 | Aliens | -0.17 | .05 (-0.19) | 0.29 (.51) | 0.31 (.48) | 0.30 (.48) | - | | | | | | | |
| 7 | Dinosaurs | -0.11 | .07 (0.51) | 0.17 (0.07) | 0.08 (0.03) | 0.08 (.28) | 0.14 (0.18) | - | | | | | | |
| 8 | Ghosts | -0.07 | -0.15 (0.17) | 0.16 (.36) | 0.14 (0.28) | 0.15 (.36) | 0.48 (.51) | 0.02 (.32) | - | | | | | |
| 9 | Unicorns | -0.12 | -.19 (0.01) | 0.11 (.55) | 0.22 (.49) | 0.19 (.57) | 0.38 (.59) | -.28 (0.28) | 0.41 (.42) | - | | | | |
| 10 | Dragons | 0.09 | .25 (0.05) | 0.16 (0.26) | 0.13 (.43) | 0.14 (.40) | 0.35 (.58) | 0.04 (0.31) | 0.44 (0.45) | 0.50 (0.67) | - | | | |
| 11 | Spongebob | -0.05 | .03 (0.3) | 0.33 (.42) | 0.34 (0.27) | 0.37 (0.35) | 0.35 (0.34) | 0.19 (0.17) | 0.55 (0.29) | 0.35 (0.70) | 0.34 (0.52) | - | | |
| 12 | Princess Elsa | -0.16 | -.02 (0.14) | 0.39 (.43) | 0.37 (0.28) | 0.39 (.45) | 0.34 (0.32) | 0.02 (0.21) | 0.57 (0.14) | 0.53 (0.85) | 0.30 (.39) | 0.69 (.74) | - | |
| 13 | Peter Pan | -0.03 | -.06 (0.2) | 0.23 (0.41) | 0.30 (0.54) | 0.30 (.46) | 0.33 (0.39) | 0.17 (0.20) | 0.40 (0.39) | 0.44 (0.85) | 0.42 (0.42) | 0.63 (.95) | 0.48 (0.79) | - |

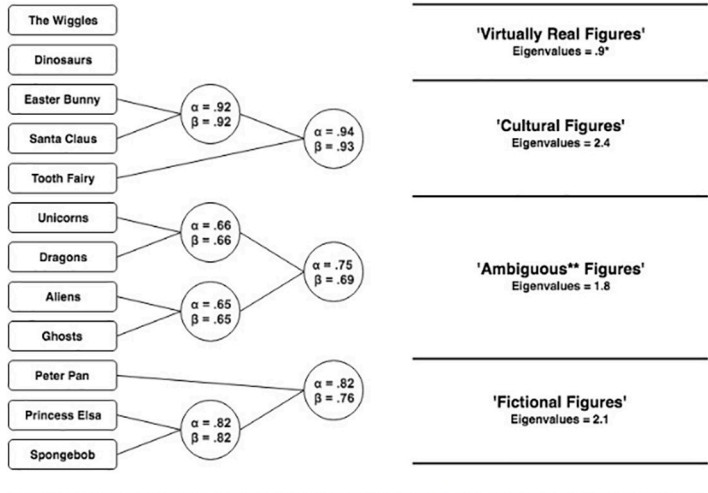

**Fig 6. Cluster analysis with a 5 cluster solution.**

they were collected under different circumstances, and with a slightly different scale), though we hope to illustrate that the same broad descriptive patterns are apparent in the replication data (also note that we used the categories in Fig 8 derived from the categories in study 1).

Per study 1, we conducted a multilevel model. However, given *a priori* analyses, we simply compared the null model to a full model which included freely estimates intercepts and slopes (i.e., Model IV). We identified that a model with a fixed y-intercept (Model I) for all categories was significantly improved by the full model with freely estimated y-intercepts and slopes (Model II), $\chi^2(6) = 224.00$, $p < .001$, The correlation between the category codes and regression intercepts $r = -.634$, indicating that as age increased endorsement (generally) decreased. This is apparent in Fig 9 and Table 7.

## Discussion of study 2

Study 2 served three purposes: First, to replicate observations made in study 1, second, to determine whether the parent-as-research-assistant model was valid, and third, to collect data with fewer missing values.

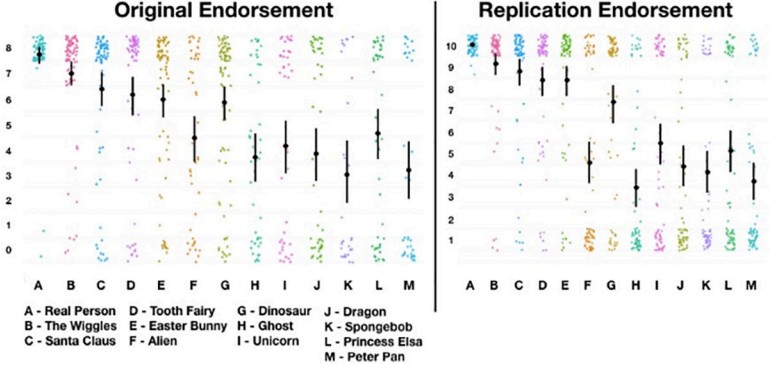

**Fig 7. Plot of original vs replication endorsement values for figures.**

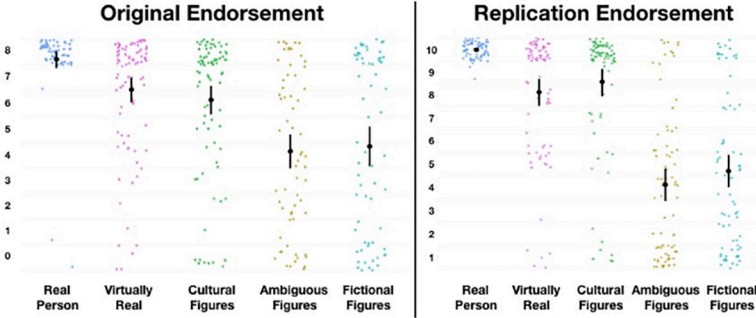

**Fig 8. Plot of original vs replication endorsement values for categories.**

The present dataset was collected by trained research assistants in a large, metropolitan science museum. Due to time constraints, we were able to collect data from 81 children (compared to 95 in our original final dataset). While study 2 did not contain individual children as young as those in study 1, we note that only very few children were younger than those in the present dataset. One unintentional difference between study 2 and study 1 was that the children in study 2 were recruited at a Science Museum. One potential implication is that these children may be more science-savvy than the children in Study 1. To what extent this influenced the data, we cannot know, yet we did observe remarkable similar observations, clusters, and model results. The most salient difference in the results of these analyses was the beta value associated with cultural figures. Specifically, in study 1 age only weakly predicts belief, while in study 2 this relationship is considerably stronger. We propose that the nature of this relationship is probably non-linear, where belief declines abruptly (and possibly totally), rather than gradually. That said, we are generally confident in our central claim that belief in a range of real and unreal figures conforms to a typology, and we predict that—beyond typical forms of testimony—that rituals and cultural norms associated with cultural figures are a special form of testimony. We hope that it is apparent that this replication, conducted by trained research assistants, has yielded results that should give any reader confidence in the method of parent-collected data and the results of Study 1.

## General discussion

To our knowledge this is the first study to examine the relative hierarchy of children's reality endorsements across a range of figures. We made several predictions. First, that figures would aggregate into a priori categories. Second, that children's endorsement would conform to a hierarchical pattern consistent with the stated criteria. Third, adults' endorsement of the

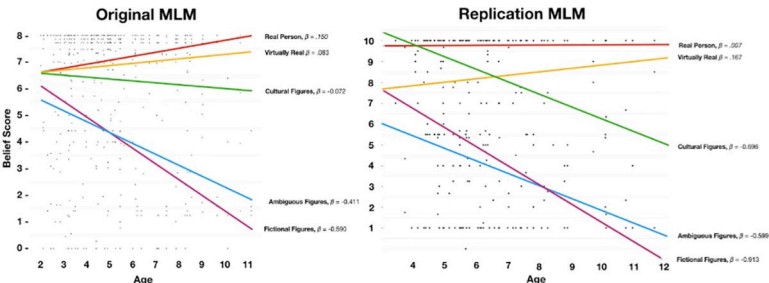

**Fig 9. Results of the multilevel model in which a child's age interacts with category of figure.**

**Table 7. Results of the regression models for determining hierarchy of belief.**

|  | Model I | Model II |
|---|---|---|
| Age (Beta; SE) | - | -.387 (0.207) |
| Constant (int; SE) | 7.020 (0.182)*** | 9.462 (1.028)*** |
| Observations | 378 | 378 |
| Log Likelihood | 1014.300 | -902.299 |

*** p< .01

realness of figures formed basic categories with less ambiguity. And fourth, age would predict endorsement patterns that converge on sensible 'adult like' endorsement. We found broad support for each hypothesis in two separate studies. We are also eager to note that the method by which we collected data—by recruiting parents online, who subsequently acted as research assistants—appears to have yielded results that do not meaningfully or systematically differ from results collected by trained research assistants.

Regarding our first hypothesis, we identified that 12 figures aggregated into four clusters which broadly corresponded to our predictions. This pattern we observed is most easily seen in Figs 1 and 2. Our second hypothesis was that children's endorsement would conform to a hierarchical pattern consistent with our stated criteria, and our fourth hypothesis was that endorsement among children would become more adult-like with age. Using multilevel analyses, we identified a clear hierarchy, one which becomes increasingly apparent among older children (i.e., the regression lines diverge at different rates, and are most different amongst the oldest age group; In technical sense, these differential values are associated with a greater model fit when we freely estimated the slopes of each categories relative to a model in which we constrained the slopes to a single regression coefficient). Our third hypothesis, that adults would have three categories of figures—real, not-real, and ambiguous—was also supported.

Regarding the interaction between categories of figures and age: The biggest deviation in our results from our hypothesis was that dinosaurs and aliens did not aggregate. But rather, dinosaurs aggregated with The Wiggles in study 1 and that dinosaurs and wiggles did not aggregate in study 2. We termed this cluster 'virtually real' as children have had no direct evidence of either figure, and yet both figures are/were real—presumably according to different criteria. Children appear to recognize that the wiggles are extant, and that dinosaurs were once extant. Indeed, in study 1 we observed that the difference in endorsement scores between Dinosaurs (and Aliens) was not significant between children and adults. This suggests that children (as a group) appear to have an already stable and adult-like understanding of Dinosaurs (and Aliens).

We had hypothesized the existence of a 'mythical' cluster—of unicorns, dragons, and ghosts—which did not materialise (though we do note that dragons and ghosts did not differ in levels of endorsement (Study 1: $p$ = .351; Study 2, $p$ = .066) that dragons did not differ from unicorns (Study 1: $p$ = .909; Study 2: $p$ = .091) and that unicorns did not differ from ghosts (Study 1: $p$ = .057). (In study 2 the difference was significant, $t(58)$ = 2.955, $p$ = .005, such that Unicorns, M = 5.45, were more endorsed than Ghosts, M = 3.406). Rather, unicorns clustered with fictional figures (in study 1), and with dragons, aliens and ghosts (in study 2); while dragons and ghosts clustered with aliens (in both studies). We note that Adults tended to endorse aliens and ghosts at relatively high levels (relative to each other, and other figures), but did not strongly endorse unicorns. Indeed, we were surprised at the levels adults endorsed ghosts. Children endorse ghosts significantly more than adults, but only by a small margin. And yet, aliens, ghosts, and dragons all provide the same amount of evidence for their own existence,

make no demands on believers (inasmuch as behavior is concerned) and are all idiosyncratically presented in media (as evil, friendly, benevolent, greedy, generous, and so on). Any attempted explanation of this would be post hoc speculation, and so we hereby refrain, and look forward to future research efforts. The fictional cluster needs little discussion, except to say that age predicts declining endorsement more strongly for this category than any other.

As predicted, we found that Santa, the Tooth Fairy, and the Easter Bunny aggregated into the *a priori* category of 'Cultural Figures' in both studies. We believe that this is primarily a consequence of cultural rituals, as outlined in the introduction. At least at the population level, it is the case that many families practice acts associated with these figures, and that these acts are relatively standardized across the group. While the present analysis is not sufficiently controlled to quantify the influence that such rituals and behaviors play, we note that the decline in endorsement for cultural figures is considerably less steep than the decline in endorsement for ambiguous and fictional figures in study 1, but is roughly as steep as it is ambiguous figures in study 2. While our model assumes a linear relationship between age and belief for these figures, we suspect it is actually non-linear in real life (a point to be examined by future research). An additional point of discussion is that children are likely exposed to more information about Santa than the Tooth Fairy or the Easter Bunny. Though it's largely an empirical question as to how much exposure children receive for all the figures in studies, we maintain that Cultural Figures are qualitatively different from other figures—particularly their nearest-endorsed-neighbors, aliens and dinosaurs—as they are coupled with rituals and behavioral norms.

The results of our multilevel model expands on existing findings, which has shown that 'more than half of 8 year olds are still in transition [to disbelief in santa]' [7] and that most, but not all, 9 year olds have abandoned belief in Santa [8]. Blair, McKee, & Jernigan [38] report similar findings. Our MLM shows that there appears to something interesting happening between the ages of 7 and 9 with regard to declining endorsement, such that the figures which we predicted to be least endorsed assume the lowest rank in the pantheon, while it is in this period that 'virtually real' figures (such as dinosaurs and humans appearing on television) appear to assume a higher standing than that of cultural figures (primarily observable in study 2). Speculatively, we believe this relationship is more pronounced than described by our models: we suspect that belief does not decline linearly (as our models assume) but may in fact be non-linear, such that after a critical threshold is crossed, belief dissipates. That said, only future research, using longitudinal methods and non-linear modelling may resolve this question. We leave the point open to future researchers.

Our hypothesized structure was based primarily on the following features: 1) direct evidence, 2) indirect evidence, 3) ritual and behavioral norms associated with the figure, and 4) explicit presentation as fiction. We tentatively argue that behaviors associated with figures—specifically, cultural rituals and behavioral norms—are a key feature that supports the endorsement of cultural figures at rates approaching that of real figures (conceding explicitly that this was not an experimental design). Of course, we can only make inferences to this causal factor in the present work, and we have no data to show what kind of behavioral commitments were apparent in the lives of the children we studied. (Though we have work in preparation in which this was the focus).

We should note, however, that we are not attempting to dismiss the role that various kinds of testimony play [30], nor the particular language used in such testimony [39], nor the source [39,40]. Rather, we would like to elevate in significance the potential role that cultural rituals and behavior play as a source of evidence alongside these well established features (keeping in mind that participation in ritual has been described previously as a special kind of testimony, albeit theoretically rather than empirically;[13]). Our intention was to conduct longitudinal

analyses in order to test this hypothesis more directly (see S1 Data), but were scuttled by issues associated with missing data (in lieu of these analyses we have presented descriptive statistics in Supplementary material A).

We would also like to emphasise the quality of the data collected in Study 1—where parents administered the survey to their children—as evinced by its similarity to the data collection in Study 2. We found no systematic or meaningful differences in our data (save for the fact that RA collected data contained fewer missing values). We broadly replicated the factor structure, multilevel model, and hierarchy within this data (as well as finding strong age effects for each category in linear regression models). We hope that this replication attempt bolsters support for the present findings, and—more broadly—reveals the value of using parents as research assistants: provided sufficient guidelines and briefing, it appears that parents may be able to efficiently and honestly report the belief, opinions, and preferences of their children, at least to a degree comparable with a research assistant unknown to the child. The extent to which either method (parent vs. RA) introduces idiosyncratic biases is unknown, but the net-results suggest that neither method is empirically inferior to the other in the present context (save for the advantage that RA's appear to generate less missing data—though this may be due to the nature of the topic, rather than a function of data collection). Largely, we believe that the parent-collection model may be an efficient way to collect data of acceptable quality, even on topics that might be intuitively considered difficult (such as belief in figures like Santa).

We believe there are three primary limitations, and some additional, smaller limitations. The first primary limitation, is that children may 'meet' some figures in real life, thus having something like direct evidence for the figure; second, the emotional impact and valence of the figures likely plays a role, and this was not accounted for; and third, what exactly does 'real' mean in this context. This work should be considered a first attempt at demonstrating the *presence* of a hierarchy (rather than a dissection of stated phenomenon), and a sincere advancement of the hypothesis (stated by authors elsewhere) that rituals play an important role in facilitating belief in non-natural figures. We also advance the *hypothesis* that rituals are a special form of testimony, rather than confidently asserting it is a known and quantified *explanation*.

Regarding direct evidence for non-natural figures: nearly 41% of all children who responded to questions about Santa reported having seen him in real life (the same proportion as those who claim to have seen the Wiggles; see S1A-S1B Tables in S1 Data). Though we argue that cultural figures do not routinely provide *direct* evidence of their own existence, but rather that they provide *indirect* evidence of their existence (i.e., gifts, chocolate, and money, respectively). We do not at presently know how many children would report having seen the Easter Bunny or the Tooth Fairy, but the extant literature suggests that the Santa data is not surprising. At least in the United States (noting that our sample was Australian) children often visit a 'live Santa Claus' in shopping centres and malls, with many visiting two or more live Santas per year [14]. In these situations, children get to interact and physically be in the presence of a cultural figure, promoted as "*the real thing*". This said, that the Tooth Fairy and The Easter Bunny are endorsed at comparable levels to Santa suggests that either direct evidence is not that important (which is highly unlikely), or that ritual participation is surprisingly powerful. Similarly, it is also somewhat popular for actors to dress as Princess Elsa as various kinds of events, and so it is possible that some number of children have also 'met' Princess Elsa. However, this practice is somewhat distinct in that children are less likely to seek out Elsa, and do not engage in a typical and predictable set of culturally dictated customs (sitting on Santa's knee, describing oneself as good or bad, and making a wish for a present).

With regard to emotional valence and impact: it is likely that the concrete positive associations of the cultural figures contributes to a child's endorsement of these figures—in classic

learning terms, Santa, the Easter Bunny, and the Tooth fairy provide positive reinforcement for a position favourable to their existence, and it's hard to make such a claim for any other figure in our pantheon. Moreover, the figures in our list all have positive associations, even those that may also be represented as negative (e.g., dragons, ghosts). We did not, for example, include more classically negative figures like vampires, or zombies (nor more esoteric but undeniably 'evil' figures like succubi or lich). While there were ethical reasons for us to not include such figures, we speculate that 'bad is stronger than good' [41], and acknowledge that there is some evidence that 'scary' stimuli are more easily represented by children than non-scary stimuli [4], though this effect fails to replicate in a similarly powered (albeit underpowered) study [42]. Though Kapitány et al. (under review; preprint available) describe the short-comings of empirically inculcating any kind of supernatural belief, and so while we believe it's possible (and even likely) emotional valence and intensity play a role, we cannot conclusively draw on any empirical data.

Finally, an important criticism for this work, as with similar research, is what exactly we mean by the term *'real'*, what exactly children understood the term to mean [43,44], and the nature of the measurement. These questions are legitimate, and a full discussion is beyond the scope of the present enquiry and the philosophical expertise of the authors. However, we understand the term 'real' here to correspond with *'confidence in the claim that [the figure] exists or has existed in some embodied and autonomous way'*. That we find middling rates of endorsement for a number of agents (among all populations studied) indicates not that something is real and unreal at the same time, but that one has a *middling-confidence* that a thing is real. One potentially problematic aspect of this can be highlighted by adults' responses to 'Dinosaurs'. There was a small number of floor responses for dinosaurs, which may reflect a belief that dinosaurs—though once real—are no longer real. However, some small proportion of those responses may also reflect the beliefs of religious fundamentalists. Largely, however, we think these floor responses—though small in number—are legitimate, and should in aggregate be taken to represent the diversity of respondents understandings of the term 'real' (indeed, some children anecdotally reported that dinosaurs are *no longer* real, which appears to us to represent a relatively sophisticated understanding of the term, and should be taken into account when the same children report that Santa *is* real). We also note that the measurement we used—particularly for fictional figures—tended to cluster at the floor and the ceiling producing a potentially misleadings 'mean' value. While regrettable, we note that the same problem does not occur for the ambiguous and cultural figures, which tend to reveal middling endorsements *by individuals* (in the former case; as revealed by the scatter plots of individual values in Fig 7), and high confidence in their existence in the latter. During the peer-review process, a reviewer suggested that rather than the multiple categories we propose, it may be more sensible—based on an interpretation of mean values for *category* data (see Fig 8)—to suggest only three categories: *real*, *less real*, and *doubtful*. We are not opposed to a 'clumping' strategy over a 'splitting' when making future hypotheses about this proposed pantheon. The question all researchers ought to ask is: what utility do my distinctions serve? In the present case, we had rather fine-grained hypotheses about the structure of the pantheon, and wished to report and interpret at a corresponding level. Certainly three, rather than five, categories may serve some purposes particularly well, though we hope that our data—imperfect that it is—can serve as a foundation for future theory-building based on the relative contribution of different forms of evidence and testimony, as described in brief by Woolley and Ghossainy [13]. Future research may examine the nature of these differences and similarities using alternative statistical and theoretical models, though we again note that our data is publicly available for the interested reader, and we are open to future discussion and collaboration on this topic. We maintain, however, that our results, measurement, and analyses suggest that children are

generally capable of admitting nuance and ambiguity into their reality endorsements, and this is apparent at population levels. That said, that we replicated our results in a second study using traditional methods of enquiry should provide some confidence that the question is reliable and face-valid.

In anticipating some critique, one might question how sensible it is to include 3-year olds in the present dataset. Our original intention was to examine longitudinal differences, and so to observe the degree to which children vary over the course of the year, and in response to cultural events. While we were unable to run these analyses, that's not to suggest the inclusion of the youngest might not yield meaningful aggregate data. Indeed, if the youngest have the greatest noise to signal ratio, it doesn't necessarily skew our results, and may more accurately inform us of the nature of changing beliefs during the earliest years. That said, we re-examined all analyses and excluded children 3-years and younger, and found no meaningful or statistical differences in our results (primarily because there were so few that young in our sample).

The data we present here suggests that, when children (and even adults) are asked to report their subjective confidence that various kinds of real and non-real figures exists, they are not only able to do so, but do so with nuance, and in a consistent pattern apparent at the population level. We argue that the possibility that cultural rituals and normative requirements on the part of figures like Santa, the Easter Bunny, and the Tooth Fairy may be very powerful cultural tools that lead to children believing such figures are real. And we anticipate—but cannot address directly—that ritual involvement is a key determinant in adult beliefs for more institutional supernatural figures such as the deities of major world religions. We hope that treating the corpus of real, supernatural, and fictional figures as a kind of *pantheon*, united by a coherent structure of underlying qualities, opens the door for higher resolution understanding of how children come to understand what is real and what is not (even when they are wrong), and allow for more nuanced approaches to research when investigating the predictors of belief and endorsement. In addition to testimony, content, and source information, we must pay more attention to the specific role that cultural rituals play in widespread belief of culture-bound supernatural figures.

## Supporting information

**S1 Fig. Plots of endorsement for Time 2 (Early September, 2015).**
(TIF)

**S2 Fig. Plots of endorsement for Time 3 (Early November, 2015).**
(TIF)

**S3 Fig. Plots of endorsement for Time 4 (Mid December, 2015).**
(TIF)

**S4 Fig. Plots of endorsement for Time 5 (Mid February, 2016).**
(TIF)

**S5 Fig. Plots of endorsement for Time 6 (Early April, 2016).**
(TIF)

**S6 Fig. Plots of endorsement for Time 7 (Late May, 2016).**
(TIF)

**S1 Appendix. Parents agreed to the following conditions before helping their child complete the survey.**
(DOCX)

**S1 Data. A note on data collection and analyses.**
(DOCX)

**S2 Data. Training questions.**
(DOCX)

**S1 Table. Responses to epistemological question for four figures.**
(DOCX)

## Acknowledgments

We like to thank Brisbane's Science Museum for supporting this research. We thank the children and parents who participated in this research.

## Author Contributions

**Conceptualization:** Rohan Kapitány, Nicole Nelson, Thalia R. Goldstein.

**Data curation:** Rohan Kapitány.

**Formal analysis:** Rohan Kapitány.

**Investigation:** Rohan Kapitány, Nicole Nelson.

**Methodology:** Rohan Kapitány, Nicole Nelson.

**Project administration:** Rohan Kapitány, Nicole Nelson.

**Writing – original draft:** Rohan Kapitány, Nicole Nelson, Emily R. R. Burdett, Thalia R. Goldstein.

**Writing – review & editing:** Rohan Kapitány, Nicole Nelson, Emily R. R. Burdett, Thalia R. Goldstein.

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
