## [Decision Letter · Decision Letter 0]

25 Feb 2020

PONE-D-20-01141

The Child’s Pantheon: Children’s Hierarchical Belief Structure in Real and Non-Real Figures

PLOS ONE

Dear Dr Kapitany,

Thank you for submitting your manuscript to PLOS ONE. After careful consideration, we feel that it has merit but does not fully meet PLOS ONE’s publication criteria as it currently stands. Therefore, we invite you to submit a revised version of the manuscript that addresses the points raised during the review process.

There are two main areas of contention:

1. The reliability of the method, and especially in the ability of children to understand degrees of probable reality. Speak to the plausibility of the assumption that children understand the question and can use the scale appropriately, either with some validation data, some commentary on your own results, or previous research.

2. The interpretation of your findings. Both reviewers took exception to those, and their criticisms should be addressed both in the text and in your cover letter. 

We would appreciate receiving your revised manuscript by Apr 10 2020 11:59PM. I am likely to invite at least one of the two reviewers back to examine the resubmission. 

To enhance the reproducibility of your results, we recommend that if applicable you deposit your laboratory protocols in protocols.io, where a protocol can be assigned its own identifier (DOI) such that it can be cited independently in the future. For instructions see: http://journals.plos.org/plosone/s/submission-guidelines#loc-laboratory-protocols

We look forward to receiving your revised manuscript.

Kind regards,

Jonathan Jong, PhD

Academic Editor

PLOS ONE

Journal Requirements:

2. Please do not include funding sources in the Acknowledgments or anywhere else in the manuscript file. Funding information should only be entered in the financial disclosure section of the submission system. https://journals.plos.org/plosone/s/submission-guidelines#loc-acknowledgments

3. Please provide additional details regarding participant consent. In the ethics statement in the Methods and online submission information, please ensure that you have specified what type of parental consent you obtained (for instance, written or verbal, and if verbal, how it was documented and witnessed). If verbal, please specify: 1) whether the ethics committee approved the verbal consent procedure, 2) why written consent could not be obtained, and 3) how verbal consent was recorded.

Reviewers' comments:

Reviewer's Responses to Questions

**Comments to the Author**

1. Is the manuscript technically sound, and do the data support the conclusions?

Reviewer #1: Partly

Reviewer #2: Yes

2. Has the statistical analysis been performed appropriately and rigorously? 

Reviewer #1: I Don't Know

Reviewer #2: Yes

3. Have the authors made all data underlying the findings in their manuscript fully available?

Reviewer #1: Yes

Reviewer #2: Yes

4. Is the manuscript presented in an intelligible fashion and written in standard English?

Reviewer #1: Yes

Reviewer #2: Yes

5. Review Comments to the Author

Reviewer #1: This is an interesting and thoughtful attempt to examine children's ontological beliefs in a more nuanced fashion than is customary. To simplify somewhat, a great deal of past research has assumed that children classify entities in a dichotomous fashion - as real or unreal. Instead, the authors argue, children have a more differentiated notion of reality with real people at one end, fictional characters at the other, with other categories in between.

Despite the strength of the theoretical framing I have several reservations about the data that the authors offer in support of their thesis.

1. In both studies, the authors asked a wide age range of children to situate each character on a 9-point star scale using training feedback that included the following: " If you think it's probably real, then choose a big number of stars, if you think it's probably not real you should choose a smaller number of stars". I think the assumption that children, especially preschool children, can understand varying degrees of probability and translate them into a varying number of stars is quite dubious. The fact that similar results were obtained whether or not parents served as the interviewer does not show that children understood the scale. For all we know, children were subject to the same misunderstanding or partial understanding of the response scale no matter who interviewed them.

2. Eyeballing the data in Figure 5 (Endorsement for categories), there seems to be little justification for claiming that children differentiate between virtually real characters (e.g. The Wiggles) and cultural figures (e.g. Santa Claus) or between Ambiguous figures (e.g. ghosts) and Fictional Figures (e.g. unicorns). A more cautious and defensible conclusion might be that children make a 3-way split between real figures, less real figures and doubtful figures.

3. The authors elaborate on various ways in which different categories of figure might be distinguished (see Table 2 for a listing of these criteria). However, they provide no evidence that these criteria were deployed by children in making their reality judgements. In particular, the authors did not ask children to offer a justification for their judgments even though such justifications might have lent support to the authors' claims. In the absence of such justification data, the authors have to rely on the reality judgment data alone and yet there are all sorts of reasons why children might make different reality judgments for different figures. For example, Figure 3 suggests that children were less confident of the existence of Peter Pan than the Tooth Fairy. This might reflect the fact that children receive indirect evidence for the Tooth Fairy and not for Peter Pan (as implied by the criteria set out in Table 2) but another equally plausible possibility is that children hear more references to the Tooth Fairy (and from more people) as compared to references to Peter Pan.

4. I am sympathetic to the authors' proposal that the rituals and norms associated with certain figures might constitute a form of testimony. However, I doubt whether a non-experimental study in which figures that are presumed to be associated with rituals and norms are compared with figures that are presumed to not be so associated are compared can provide strong evidence for their claim. Those two classes of figure will be very likely to vary in other ways (see comment #3 above), weakening any conclusions that can be drawn.

Reviewer #2: This paper presents the results of two surveys of children’s and adults’ beliefs about a range of real and fantastical beings. Specifically, the authors presented participants with various beings that they had categorized a priori into 5 groups as a function of a number of characteristics and attributes that they reasoned would affect judgments about reality. Participants were asked to indicate how real they thought the entities were, using a Likert scale. Results indicate that children and adults exhibit different patterns of judgments regarding these entities, and are interpreted as revealing that children’s judgments reflect the predicted characteristics/attributes of the beings, and specifically the important role of participation in the behavioral rituals surrounding the entities.

This study is unique in attempting to explore a more nuanced representation of reality than in previous studies of this topic, many of which have simply assessed beliefs about whether something is real or not real. This is a laudable goal. The focus on the role of involvement in behavioral rituals is also novel and important. In general, I really like that the authors are trying to determine how various factors like ritual participants and evidence interact to engender belief.

I should note that I reviewed an earlier version of this paper for another journal. I mention this, because I find this version significantly improved over the original version, and that affects my evaluation (in a positive way). Overall, though, I still have some concerns about the paper, and they follow.

1. On p. 20, the authors discuss whether endorsement becomes more adult-like with age. The text states that, “it is clear from the data presented in Figure 4 that the negative slopes associated with fictional characters and cultural characters..” From the graph that appears to me to be Figure 4, it doesn’t look like there’s any decline at all in belief in cultural figures—that line looks flat. First, this doesn’t fit with the text, and second, there’s so much research showing that these beliefs decline with age that something must be odd about these data. Can the authors clarify?

2. As the authors note, in terms of the hierarchy, for the most part the data from Study 2 replicate Study 1. But I was struck by the differences in the correlations. For the most part, they’re lower in Study 2. Could the authors speculate on why this might be the case?

3. Another difference between the two studies is that in Study 2, the slope for Cultural Figures actually does appear to be negative. Why this difference? The authors make much in the General Discussion regarding how the Study 2 results validate the method used in Study 1 but this is a critical difference. Apparently in an attempt to explain it, they state that it is probably due to belief declining abruptly rather than gradually (this is stated both in the discussion of Study 2 and in the General Discussion). First, I don’t understand how that explains the difference. Could the authors expand upon their explanation? Second, I only know of one study that looks specifically at Santa disbelief, and that study finds that belief does not decline abruptly: As Anderson & Prentice (1994) state, “the ultimate realization came about through a gradual transition to disbelief rather than at a single point of demarcation” (p. 74).

4. On p. 28, the authors state that “endorsement for cultural, ambiguous, and fictional figures declines at differential rates” but it looks to me like in Study 2 (Figure 9, I think), the lines for cultural and ambiguous are parallel. (And, as already mentioned, in Study 1 it looks like cultural doesn’t decline). Could the authors clarify?

5. I’m a little confused about the paragraph at the bottom of p. 28 as well. The text states that dinosaurs aggregated with The Wiggles. But on p. 24 they state that Dinosaurs and Wiggles didn’t aggregate (and that in the “primary dataset this cluster had relatively low reliability”). Please clarify.

6. On p. 30 the authors state that the “most unbelievable figures” assume the lowest rank in the pantheon. What is meant by “most unbelievable figures”? They also state that ambiguous figures “assume a higher standing than that of cultural figures.” I guess I’d like the language to be more precise in both of these cases. Also, it doesn’t actually seem to be true. In Study 2, it looks like fictitious and ambiguous features are changing their relative standing and in Study 1 there is no change in standing (again, I’m looking at what I think is Figure 9).

7. On p. 33 the authors seem to be discussing individual subject patterns but I didn’t see any discussion or analysis of these in the Results sections. The authors should include these data if they want to discuss them.

8. There are numerous grammatical errors through the manuscript. I recommend a very close checking before submitting a revision.

Some minor points:

1. P. 10, states that “endorsement between these figures is not statistically different.” First, the authors needs to specific to which figures they are referring, that is, between Santa and the other two, or between the other two? Second, a citation needs to be provided for this claim.

2. P. 10, what is meant by the “four factors of testimony”?

3. For Study 1, 59 out of 154 children’s data were dropped. Were these all due to not answering the last question? I’d like a little more information about this and its implications regarding the generalizability of the findings.

4. Pp. 14-20, the “hypotheses” in the subheadings aren’t hypotheses. I’d suggest rephrasing.

5. P. 20 second paragraph, author should indicate explicitly that these data concern the “Ambiguous” category members.

6. None of the (main) figures in my copy are labeled so I had to guess figure numbers.

7. What’s wrong with including 3-year-olds (p. 33)? Lots of studies on this topic do so.

6. PLOS authors have the option to publish the peer review history of their article (what does this mean?). If published, this will include your full peer review and any attached files.

Reviewer #1: No

Reviewer #2: No

---

## [Author Response · Author response to Decision Letter 0]

8 Apr 2020

Please see attached 'response to reviewers'. The response is far too wordy to include here.

---

## [Decision Letter · Decision Letter 1]

20 May 2020

The Child’s Pantheon: Children’s Hierarchical Belief Structure in Real and Non-Real Figures

PONE-D-20-01141R1

Dear Dr. Kapitany,

We are pleased to inform you that your manuscript has been judged scientifically suitable for publication and will be formally accepted for publication once it complies with all outstanding technical requirements.

With kind regards,

Jonathan Jong, PhD

Academic Editor

PLOS ONE

Additional Editor Comments (optional):

Reviewers' comments:

Reviewer's Responses to Questions

**Comments to the Author**

1. If the authors have adequately addressed your comments raised in a previous round of review and you feel that this manuscript is now acceptable for publication, you may indicate that here to bypass the “Comments to the Author” section, enter your conflict of interest statement in the “Confidential to Editor” section, and submit your "Accept" recommendation.

Reviewer #2: All comments have been addressed

2. Is the manuscript technically sound, and do the data support the conclusions?

Reviewer #2: Yes

3. Has the statistical analysis been performed appropriately and rigorously? 

Reviewer #2: Yes

4. Have the authors made all data underlying the findings in their manuscript fully available?

Reviewer #2: Yes

5. Is the manuscript presented in an intelligible fashion and written in standard English?

Reviewer #2: Yes

6. Review Comments to the Author

Reviewer #2: (No Response)

7. PLOS authors have the option to publish the peer review history of their article (what does this mean?). If published, this will include your full peer review and any attached files.

Reviewer #2: Yes: Jacqueline D Woolley

---

## [Editor Report · Acceptance letter]

26 May 2020

PONE-D-20-01141R1 

The Child’s Pantheon: Children’s Hierarchical Belief Structure in Real and Non-Real Figures 

Dear Dr. Kapitány:

I am pleased to inform you that your manuscript has been deemed suitable for publication in PLOS ONE. Congratulations! Your manuscript is now with our production department. 

With kind regards,

on behalf of

Dr. Jonathan Jong 

Academic Editor

PLOS ONE